# Prevalence, trends, and factors associated with maternal autonomy regarding healthcare, finances, and mobility in Bangladesh: Analysis of Demographic and Health Surveys 1999–2018

**Gulam Muhammed Al Kibria**[1]*, **Jennifer Albrecht**[2], **Wendy Lane**[2], **Kristen A. Stafford**[2], **Laundette Jones**[2], **Roumen Vesselinov**[2], **Jon Mark Hirshon**[2,3,4]

1 Department of International Health, Johns Hopkins Bloomberg School of Public Health, Baltimore, Maryland, United States of America, 2 Department of Epidemiology and Public Health, University of Maryland School of Medicine, Baltimore, Maryland, United States of America, 3 Department of Emergency Medicine, University of Maryland School of Medicine, Baltimore, Maryland, United States of America, 4 Baltimore VA Medical Center, Baltimore, Maryland, United States of America

* gkibria1@jhu.edu

## Abstract

Maternal autonomy is associated with improved healthcare utilization/outcomes for mothers and babies in low- and middle-income countries. We investigated the trends in the prevalence and factors associated with maternal autonomy in Bangladesh. This cross-sectional study analyzed the Bangladesh Demographic and Health Survey for 1999–00, 2004, 2007, 2011, 2014, and 2017–18. Maternal autonomy was defined as at least one decision-making ability regarding healthcare, large household purchases, and freedom of mobility. We included 15-49-year-old mothers with at least one live-birth in the past three years. We compared the samples based on the presence of autonomy and reported the trends in prevalence (95% confidence intervals (CIs)) across the survey years. Lastly, we performed multilevel logistic regression to report prevalence odds ratios (PORs) for the associated factors. Variables investigated as potential factors included maternal age, number of children, maternal education, paternal education, current work, religion, mass media exposure, wealth quintile, place and division of residence, and survey years. The prevalence of 'any' maternal autonomy was 72.0% (95% CI: 70.5–73.5) in 1999–00 and increased to 83.8% (95% CI: 82.7–84.9) in 2017–18. In adjusted analysis, mothers with older age, higher education, work outside the home, and mass media exposure had higher odds of autonomy than their counterparts (POR > 1, p < 0.05). For instance, compared to mothers without any formal education, the odds of autonomy were significantly (p < 0.001) higher among mothers with primary (adjusted POR: 1.2, 95% CI: 1.1–1.4), secondary (adjusted POR: 1.4, 95% CI: 1.2–1.6), and college/above (adjusted POR: 1.9, 95% CI: 1.6–2.2) education. While the level of maternal autonomy has increased, a substantial proportion still do not have autonomy. Expanding educational and earning opportunities may increase maternal autonomy. Further research should investigate other ways to improve it as well.

**Data Availability Statement:** Data is available to download upon approval from: https://dhsprogram.com/data/available-datasets.cfm

**Funding:** The authors received no specific funding for this work.

**Competing interests:** The authors have declared that no competing interests exist.

## Introduction

Maternal autonomy or decision-making ability about families indicates overall socioeconomic development. It is often measured as a combination of several dimensions of decision-making ability regarding healthcare, finances, and mobility. It reflects the ability and self-confidence of women to overcome external barriers through their interaction with husbands, family members, and other stakeholders [1–3]. Studies from Bangladesh and other low- and middle-income countries (LMICs) have reported that maternal autonomy is associated with improved healthcare utilization (e.g., antenatal care visits, institutional delivery, and postnatal care) and outcomes (e.g., low birth weight) for mothers and children [4–6].

In recent decades, significant socioeconomic developments have occurred in Bangladesh, including increases in education level and women's employment [7–9]. Studies from other countries suggest that improvement in socioeconomic status (e.g., education or income) could positively impact women's autonomy [10–13]. However, the trends in the prevalence of maternal autonomy or factors associated with maternal autonomy have not been studied in Bangladesh. Considering the significance of maternal autonomy in improving healthcare utilization and outcomes of mothers and children, examining factors associated with maternal autonomy across survey years could help identify women with low autonomy who might benefit from interventions to improve autonomy.

To capture data on women's empowerment in Bangladesh, BDHS reports the level of women's autonomy during each survey cycle [7]. However, using different definitions across survey periods yields different estimates of maternal autonomy. For instance, in BDHS 2017–18, women's autonomy was reported as women's decision-making ability regarding large household purchases, healthcare, and freedom of mobility [7]. According to that report, approximately 60% of women aged 15–49 reported making these decisions alone or jointly with their husbands [7]. In BDHS 2011 and 2014, an additional variable (i.e., decision regarding child healthcare) was included, and it was found that 43% and 42% of women had autonomy, respectively [14, 15]. The number of variables used to define autonomy was higher in 2004 and 1999–00 [16, 17]. To track trends in the prevalence of maternal autonomy, using a standard definition across all survey years is essential.

The objectives of the present study are to investigate the trends in prevalence and factors associated with maternal autonomy using a standard definition from a series of nationally representative surveys in Bangladesh (i.e., 6). Based on previous reports, we hypothesize that the prevalence of maternal autonomy has increased and that women with higher socioeconomic status (e.g., education or wealth) have higher autonomy than their counterparts.

## Methods

### Data source

In this cross-sectional study, we analyzed data from the BDHS for years 1999–00, 2004, 2007, 2011, 2014, and 2017–18. BDHS is a nationally representative household survey conducted every 3–4 years in Bangladesh. It used a two-stage stratified sampling method. Enumeration areas (EAs), also known as primary sampling units (PSUs), were selected in the first stage based on the urban-rural distribution (i.e., probability proportional to size), followed by a systematic sample of households per EA in the next stage (S1 Fig). Sample sizes, target households, and completed interviews varied across survey years. Response rates ranged from 97–99%. This

study used data from the women's questionnaire administered to women of reproductive age (i.e., 15-49-year-olds) who were ever married. To conform to the BDHS methods, we included women who had given birth within three years preceding the survey [6, 14–17, 24].

### Ethics statement

The institutional review board (IRB) of the University of Maryland, Baltimore determined that this study meets the definition of non-human subject research and exempted it from IRB oversight (Approval Number: HP-00105503). Participants provided verbal informed consent to participate in the survey; if a participant was below 18 years of age, an adult household member provided consent.

### Study variables

**Maternal autonomy.** Women's autonomy was operationalized using three questions common over all survey years:

1. Person who usually decides on the respondent's healthcare

2. Person who usually decides on major household purchases

3. Person who usually decides on visits to family and relatives

Five answer options were provided: respondent alone; respondent and husband/partner; husband/partner alone; someone else; and others [7]. Responses of making each decision 'alone' or 'jointly with husband/partner' were coded as 'have autonomy'; all other options were coded as 'does not have autonomy'. By combining the three decision-making questions, a dichotomous variable was created: no (i.e., 'no' to all 3 questions, referent) and yes (i.e., 'yes' to at least 1 question). We also categorized autonomy by number of decision-making abilities: no (i.e., 'no' to all 3 questions), low (i.e., yes to 1–2 questions), and high (i.e., yes to all 3 questions) [7, 18].

**Other variables.** To identify factors associated with maternal autonomy, the following variables were selected to examine based on published literature and data structure: maternal age (i.e., 15–19, 20–29, and 30–49 years), number of children (i.e., '1', '2–3', and '4 or more'), maternal education (i.e., no formal education, primary, secondary, and college/above), paternal education (i.e., see maternal education), current work status (i.e., yes or no), wealth quintile (i.e., stratified as poorest, poorer, middle, richer, and richest), religion (i.e., Muslim or other), mass media exposure (i.e., yes or no), rural-urban place and division of residence, and survey years [1, 4, 7, 19–24]. All the variables were based on self-report except the wealth quintile, which was based on principal component analysis of basic household construction materials and other household belongings. Previous studies reported that these factors impact women's autonomy [10–13]. S1 Table describes all the study variables.

**Statistical analysis.** First, the sociodemographic and socioeconomic characteristics of the study participants were described by survey years. Variables were reported with weighted numbers of samples (n) and percentages (%). Rao-Scott's chi-square tests tested differences in distributions across survey years. For each survey year, the weighted prevalence (and 95% confidence interval (CI)) of maternal autonomy was estimated for each factor. The trends in the prevalence of maternal autonomy over survey years by selected characteristics were tested using Cochran-Armitage trend tests.

Next, the sample was compared by the presence of autonomy (any vs. none) using a similar approach. Lastly, unadjusted and adjusted multilevel logistic regression analyses were used to identify factors associated with maternal autonomy. Variables with p-values <0.05 in unadjusted analyses were added to the multivariable model. Multicollinearity among study variables

was examined with variance inflation factor using a fake multiple linear regression model. Both unadjusted and adjusted prevalence odds ratios (POR) were reported with 95% CIs. The prevalence, trends, and associated factors of at least two decision-making abilities were examined as a secondary analysis.

Data were analyzed using Stata 15.0 (statistical software, Stata Corporation, College Station, TX, USA). The 'svy' prefix in Stata command accommodates weighted survey data and accounts for survey design, coverage, and non-response when calculating estimates [25].

## Results

### Sample characteristics

Table 1 shows the sociodemographic characteristics of the study participants by survey year. The number of included mothers from BDHS 1999–00, 2004, 2007, 2011, 2014, and 2017–18 was 3803, 3761, 3352, 4652, 4627, and 5051, respectively. Most mothers (about 80%) were younger than 30 years old; the proportion of younger (i.e., 15-19-year-olds) mothers declined between 1999–00 (22.2%) and 2017–18 (17.9%). We also observed changes in several other sociodemographic characteristics. For instance, between 1999–00 and 2017–18, the proportion of mothers with at least four or more children declined from 27.7% (n = 1053) to 12.2% (n = 617). The proportion of mothers with education at or above college increased from 4.1% (n = 155) in 1999–00 to 17.1% (n = 864) in 2017–18. Similar increases in employment, paternal education, exposure to mass media, and urban residence were observed.

### Prevalence and trends of maternal autonomy

The prevalence of maternal autonomy by survey year is shown in Fig 1. From 1999–00 to 2017–18, the overall prevalence of 'no' autonomy reduced from 28.0% (95% CI: 26.5–29.5) to 16.2% (95% CI: 15.1–17.3). The prevalence of 'high' (i.e., all 3 decisions) autonomy increased from 36.4% (95% CI: 34.8–38.1) in 1999–00 to 53.7% (95% CI: 52.1–55.2) in 2017–18.

The prevalence of autonomy about own health care increased from 50.8% (95% CI: 48.5–53.0) in 1999–00 to 72.8% (95% CI: 71.1–74.5) in 2017–18 (Fig 2). For major household purchases, it increased from 56.5% (95% CI: 54.4–58.5) in 1999–00 to 66.7% (95% CI: 65.0–68.4) in 2017–18. A similar change was observed for freedom of mobility.

Table 2 shows the prevalence and trends of maternal autonomy (i.e., at least one decision-making ability) according to selected background characteristics. Regardless of characteristics, the prevalence of autonomy increased significantly across most survey years. Overall, the prevalence of any autonomy increased from 72.0% (95% CI: 70.5–73.5) in 1999–00 to 83.8% (95% CI: 82.7–84.9) in 2017–18. We also estimated the prevalence of high decision-making abilities (S2 Table); the trends were similar to Table 2.

### Comparison of the study sample by the presence of any maternal autonomy

**Factors affecting maternal autonomy.**  In Table 3, the factors associated with having any autonomy were reported. In the adjusted analysis, overall, mothers with older age, higher parity, and more education had higher odds of autonomy. For instance, compared to mothers with no formal education, those with primary (adjusted POR: 1.2, 95% CI: 1.1–1.4, p<0.001), secondary (adjusted POR: 1.4, 95% CI: 1.2–1.6, p<0.001), or college/above (adjusted POR: 1.9, 95% CI: 1.5–2.2, p<0.001) education had higher odds of having any autonomy. On the other hand, paternal education and household wealth status were not associated with maternal autonomy. We also performed several stratified analyses based on the number of decision-making abilities to compare distributions and related factors; the associated factors were

**Table 1. Maternal sociodemographic and socioeconomic characteristics by survey year, BDHS 1999–00 to 2017–18, % (n)[1].**

| Variable | 1999–00 (n = 3803) | 2004 (n = 3761) | 2007 (n = 3352) | 2011 (n = 4652) | 2014 (n = 4627) | 2017–18 (n = 5051) | p-value |
|---|---|---|---|---|---|---|---|
| **Maternal age (in years)** | | | | | | | |
| 15–19 | 22.2 | 22.3 | 21.1 | 19.6 | 21.0 | 17.9 | <**0.001** |
| 20–29 | 56.2 | 56.6 | 60.1 | 62.9 | 59.3 | 61.1 | |
| 30–49 | 21.7 | 21.1 | 18.8 | 17.5 | 19.7 | 21.0 | |
| **Number of children** | | | | | | | |
| 1 | 28.6 | 29.5 | 34.5 | 36.1 | 39.9 | 38.2 | <**0.001** |
| 2–3 | 43.7 | 42.9 | 43.9 | 46.7 | 46.3 | 49.6 | |
| 4 or More | 27.7 | 27.5 | 21.6 | 17.1 | 13.8 | 12.2 | |
| **Maternal education level** | | | | | | | |
| No education | 45.3 | 35.5 | 23.8 | 17.6 | 14.2 | 6.3 | <**0.001** |
| Primary | 28.8 | 30.5 | 30.6 | 30.0 | 27.9 | 27.6 | |
| Secondary | 21.8 | 28.4 | 38.8 | 44.8 | 47.7 | 49.0 | |
| College/above | 4.1 | 5.6 | 6.8 | 7.5 | 10.2 | 17.1 | |
| **Paternal education level** | | | | | | | |
| No education | 43.4 | 39.2 | 33.0 | 27.6 | 23.8 | 13.7 | <**0.001** |
| Primary | 24.9 | 26.6 | 28.1 | 29.8 | 30.0 | 33.7 | |
| Secondary | 21.8 | 24.4 | 27.8 | 29.8 | 31.8 | 34.1 | |
| College/above | 9.8 | 9.8 | 11.0 | 12.8 | 14.4 | 18.5 | |
| **Current work status** | | | | | | | |
| No | 83.7 | 83.7 | 76.7 | 92.0 | 76.3 | 62.7 | <**0.001** |
| Yes | 16.3 | 16.3 | 23.3 | 8.0 | 23.7 | 37.3 | |
| **Exposure to mass media** | | | | | | | |
| Not exposed | 61.1 | 42.6 | 45.1 | 50.5 | 49.0 | 44.8 | <**0.001** |
| Exposed | 38.9 | 57.4 | 54.9 | 49.5 | 51.0 | 55.2 | |
| **Religion** | | | | | | | |
| Muslim | 89.3 | 92.5 | 91.6 | 91.2 | 91.7 | 91.9 | 0.52 |
| Other | 10.7 | 7.5 | 8.4 | 8.8 | 8.3 | 8.1 | |
| **Wealth quintile** | | | | | | | |
| Poorest | 25.6 | 24.8 | 21.1 | 22.8 | 21.7 | 20.6 | **0.046** |
| Poorer | 22.2 | 20.2 | 21.5 | 19.8 | 18.9 | 20.5 | |
| Middle | 20.0 | 20.5 | 19.2 | 19.7 | 19.1 | 19.2 | |
| Richer | 16.4 | 17.5 | 19.6 | 19.6 | 20.6 | 20.2 | |
| Richest | 15.8 | 16.9 | 18.6 | 18.1 | 19.7 | 19.5 | |
| **Place of residence** | | | | | | | |
| Urban | 16.6 | 19.9 | 21.6 | 23.0 | 26.1 | 26.8 | **0.005** |
| Rural | 83.4 | 80.1 | 78.4 | 77.0 | 73.9 | 73.2 | |
| **Division of residence** | | | | | | | |

(*Continued*)

**Table 1.** (Continued)

| Variable | 1999–00 (n = 3803) | 2004 (n = 3761) | 2007 (n = 3352) | 2011 (n = 4652) | 2014 (n = 4627) | 2017–18 (n = 5051) | p-value |
|---|---|---|---|---|---|---|---|
| Dhaka | 30.5 | 30.9 | 31.8 | 30.5 | 35.3 | 25.6 | **0.001** |
| Chittagong | 21.8 | 21.6 | 21.8 | 23.3 | 21.8 | 21.2 | |
| Rajshahi | 24 | 22.3 | 22.6 | 13.3 | 10.0 | 11.6 | |
| Khulna | 10.3 | 11.1 | 9.4 | 9.5 | 8.0 | 9.2 | |
| Barisal | 6.2 | 6.0 | 6.1 | 5.6 | 5.8 | 5.7 | |
| Sylhet | 7.3 | 8.2 | 8.3 | 7.4 | 9.3 | 7.6 | |
| Rangpur | NA | NA | NA | 10.5 | 9.7 | 10.6 | |
| Mymensingh | NA | NA | NA | NA | NA | 8.5 | |

1.Weighted column percentages and weighted numbers (Column headers), sample weights were provided with each BDHS survey

Abbreviations: BDHS: Bangladesh Demographic & Health Survey; NA: Not applicable

similar to those observed for any autonomy. We also compared the sample by the presence of maternal autonomy (S3 and S4 Tables). The factors were reported by the level of autonomy (i.e., low and high autonomy, S5 Table), these were similar to the factors reported in Table 3.

## Discussion

The prevalence of maternal autonomy increased significantly in Bangladesh over the survey years, and this change was observed regardless of most sociodemographic and socioeconomic

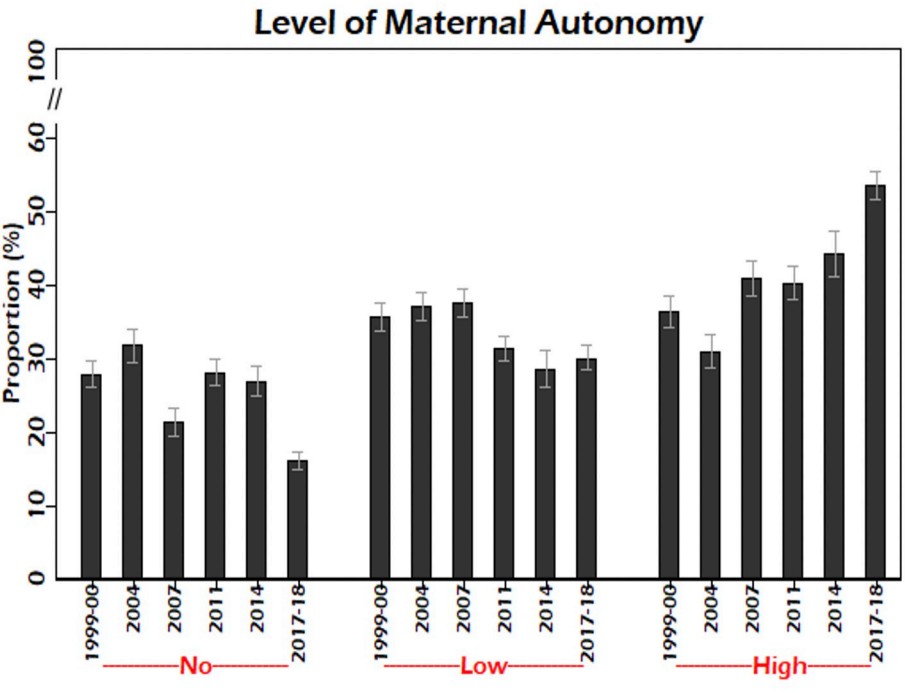

**Fig 1. Prevalence (95% CI) of maternal autonomy by survey, BDHS 1999–00 to 2017–18.** Abbreviation: BDHS: Bangladesh Demographic & Health Survey, CI: Confidence interval.

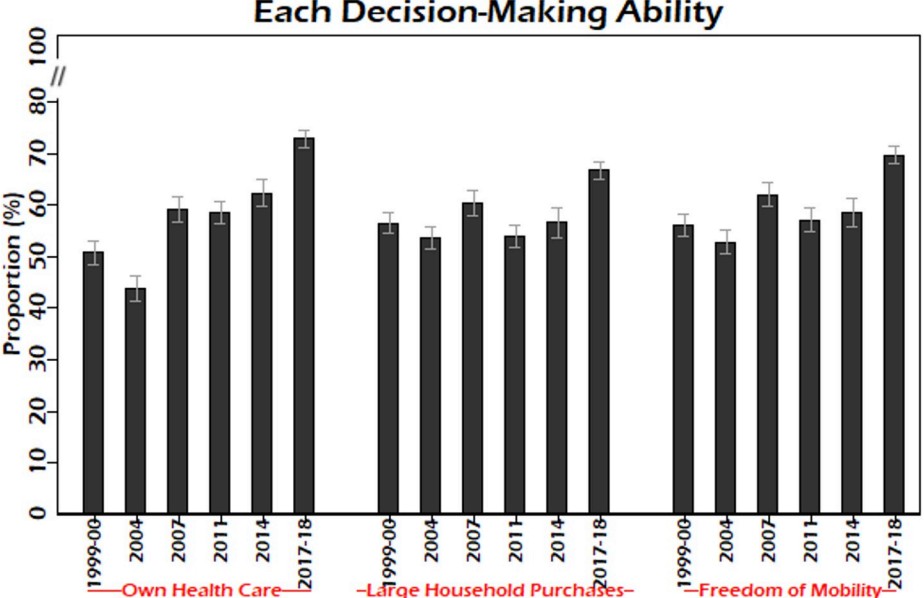

**Fig 2. Prevalence (95% CI) of each decision-making ability by survey, BDHS 1999–00 to 2017–18.** Abbreviation: BDHS: Bangladesh Demographic & Health Survey.

characteristics; this result supports our hypothesis regarding trends in the prevalence of autonomy. Older age, higher education, employment, and residence in urban regions or certain administrative divisions were associated with higher autonomy; this result also supports part of our hypothesis about the association of socioeconomic status with the outcome. The upward trend in the prevalence of maternal autonomy is a welcome finding of our study. Previous studies from other countries reported factors associated with maternal autonomy regarding health care [10–13]. Overall, those findings are consistent with our results, with mothers who are older or have higher education having a higher likelihood of having decision-making ability about healthcare compared to younger or less educated mothers.

Our findings regarding the association between maternal autonomy and age are consistent with past studies. As women get older, they develop confidence in their decision-making and problem-solving abilities, which allows them to negotiate more with their husbands about healthcare, household purchases, and mobility [11, 12]. Younger mothers are more likely to be newly married and live with their in-laws in the same household, whereas the in-laws may have more control over household decisions [10, 26].

Similar to other studies, maternal education level positively correlated with autonomy [10–13]. Education increases confidence and opportunities to overcome external barriers by decreasing the power differential between mothers and other stakeholders, including husbands, family members, and healthcare providers. Educated mothers may better understand their rights and the importance of expressing their opinions than uneducated mothers. For instance, educated women may be more aware of the value of healthcare, allowing them to talk with their husbands about healthcare [27]. They can also obtain more information from public health campaigns and other knowledge sources (e.g., social media) and identify the appropriate time to seek health care. Educated women generally have more earning opportunities, which can make them financially independent. They may also work outside the home, ultimately increasing their mobility freedom.

**Table 2. Prevalence (95% confidence interval) and trends of any maternal autonomy by sociodemographic characteristics.**

| Variable | 1999–00 | 2004 | 2007 | 2011 | 2014 | 2017–18 | Mean change |
|---|---|---|---|---|---|---|---|
| **Maternal age (in year)** | | | | | | | |
| 15–19 | 64.4 (60.8,67.8) | 60.5 (56.7,64.2) | 73.7 (69.7,77.3) | 63.8 (60.2,67.3) | 67.4 (63.5,71.1) | 70.5 (67.0,73.7) | **1.2\* (0.3, 2.1)** |
| 20–29 | 73.6 (71.6,75.5) | 68.7 (66.5,70.9) | 79.0 (76.9,81.0) | 72.2 (70.3,74.0) | 72.2 (69.8,74.5) | 84.9 (83.4,86.2) | **1.9\*\*\* (1.4,2.4)** |
| 30–49 | 75.8 (72.6,78.8) | 74.8 (71.3,78.0) | 83.0 (79.6,86.0) | 79.3 (76.0,82.2) | 81.1 (77.7,84.1) | 92.1 (90.1,93.8) | **2.9\*\*\* (2.2, 3.6)** |
| **Maternal education level** | | | | | | | |
| No education | 70.2 (67.8,72.5) | 65.8 (62.9,68.6) | 74.3 (70.7,77.7) | 67.9 (64.1,71.6) | 71.4 (66.6,75.8) | 83.1 (78.1,87.1) | **1.3\* (0.3, 2.3)** |
| Primary | 69.5 (66.5,72.3) | 69.1 (66.0,72.0) | 79.1 (76.2,81.7) | 70.5 (67.7,73.1) | 71.1 (67.6,74.4) | 84.5 (82.3,86.5) | **2.2\*\*\* (1.5, 2.9)** |
| Secondary | 76.2 (73.0,79.2) | 67.5 (64.3,70.5) | 79.4 (76.8,81.9) | 71.9 (69.7,74.0) | 73.3 (70.8,75.8) | 83.0 (81.3,84.6) | **1.7\*\*\* (1.0, 2.3)** |
| College/above | 87.7 (81.7,91.9) | 81.8 (75.6,86.7) | 87.1 (81.5,91.1) | 85.0 (80.4,88.6) | 78.5 (74.1,82.4) | 85.3 (82.5,87.7) | 0.3 (-0.3, 0.7) |
| **Paternal education level** | | | | | | | |
| No education | 67.8 (65.3,70.2) | 65.9 (63.1,68.6) | 77.2 (74.2,79.9) | 71.1 (68.1,73.9) | 72.1 (68.0,75.8) | 88.4 (85.5,90.8) | **2.8\*\*\* (2.0,3.5)** |
| Primary | 72.8 (69.6,75.7) | 68.3 (65.0,71.4) | 78.0 (74.8,80.9) | 70.3 (67.5,73.0) | 72.1 (68.7,75.2) | 85.6 (83.7,87.4) | **2.3\*\*\* (1.6, 2.9)** |
| Secondary | 74.4 (71.1,77.4) | 68.8 (65.4,72.1) | 78.3 (75.1,81.2) | 71.1 (68.3,73.7) | 72.4 (69.5,75.2) | 82.2 (80.1,84.1) | **1.5\*\*\* (0.8, 2.1)** |
| College/above | 84.2 (80.1,87.7) | 75.3 (70.3,79.7) | 85.9 (81.3,89.5) | 78.2 (74.3,81.7) | 77.4 (73.4,81.0) | 85.8 (83.1,88.2) | 0.5 (-0.3, 1.4) |
| **Current work** | | | | | | | |
| No | 70.7 (69.0,72.3) | 66.6 (64.7,68.4) | 77.3 (75.4,79.1) | 70.9 (69.3,72.4) | 71.8 (69.9,73.7) | 82.7 (81.1,84.1) | **1.9\*\*\* (1.5, 2.4)** |
| Yes | 79.1 (75.5,82.3) | 76.5 (72.4,80.1) | 83.2 (79.9,86.1) | 82.1 (77.5,86.0) | 76.6 (72.6,80.2) | 85.7 (83.9,87.4) | **1.2\*\*\* (0.5, 1.8)** |
| **Wealth quintile** | | | | | | | |
| Poorest | 67.4 (64.1,70.6) | 67.2 (63.7,70.5) | 76.2 (72.3,79.6) | 73.0 (69.8,76.0) | 66.7 (62.2,70.8) | 86.3 (84.0,88.3) | **2.6\*\*\* (1.8, 3.5)** |
| Poorer | 68.3 (64.8,71.7) | 65.5 (61.6,69.3) | 77.5 (73.8,80.7) | 70.3 (66.7,73.6) | 72.9 (69.2,76.4) | 83.0 (80.4,85.4) | **2.6\*\*\* (1.8, 3.4)** |
| Middle | 73.3 (69.8,76.5) | 67.0 (63.1,70.7) | 76.2 (72.1,79.9) | 68.3 (64.8,71.5) | 74.2 (69.8,78.2) | 82.8 (80.0,85.4) | **1.9\*\*\* (1.0, 2.7)** |
| Richer | 75.2 (71.4,78.5) | 68.6 (64.6,72.4) | 79.6 (75.7,82.9) | 71.7 (68.4,74.9) | 74.0 (70.3,77.4) | 83.4 (80.7,85.8) | **1.6\*\*\* (0.7, 2.4)** |
| Richest | 79.9 (76.6,82.8) | 73.8 (70.2,77.1) | 84.4 (81.0,87.2) | 75.6 (72.2,78.7) | 77.6 (74.2,80.7) | 83.4 (80.6,85.9) | 0.7 (-0.1, 1.5) |
| **Place of residence** | | | | | | | |
| Urban | 75.9 (73.0,78.7) | 75.5 (72.5,78.2) | 83.4 (80.8,85.6) | 78.0 (75.3,80.5) | 77.4 (74.6,80.0) | 85.8 (83.7,87.6) | **1.6\*\*\* (0.8, 2.3)** |
| Rural | 71.2 (69.5,73.0) | 66.4 (64.4,68.3) | 77.3 (75.4,79.2) | 69.9 (68.1,71.6) | 71.4 (69.2,73.5) | 83.1 (81.7,84.4) | **2.0\*\*\* (1.5, 2.5)** |
| **Overall** | 72.0 (70.5,73.5) | 68.2 (66.5,69.8) | 78.6 (77.0,80.2) | 71.8 (70.3,73.2) | 73.0 (71.2,74.6) | 83.8 (82.7,84.9) | **2.0\*\*\* (1.6, 2.4)** |

\*: p<0.05

\*\*: p<0.01

\*\*\*: p<0.001

Mothers in urban regions and certain administrative divisions had significantly higher autonomy than those in rural areas [12]. In addition to individual (e.g., age) determinants, community-level determinants such as sociocultural/socioeconomic differences between regions in Bangladesh may play a role [28, 29]. Urban residence may provide better opportunities for work, healthcare, and travel, which can decrease women's reliance on others [7]. For example, the shorter distance between homes and hospitals in urban regions than rural areas allow women to seek healthcare independently. Urban regions often have better transportation facilities than rural regions, which can reduce transportation barriers.

Additionally, urban residents are often more educated than rural residents, creating a sociocultural environment that allows women more decision-making power [7, 30]. Earlier reports have shown socioeconomic (e.g., education level) and sociocultural differences by geographic divisions, which can also contribute to divisional differences in autonomy [7, 14, 31]. These differences may also contribute to disparities in healthcare utilization and outcomes between rural-urban places and divisions in Bangladesh. There may be intra-divisional differences due

**Table 3. Factors associated with any maternal autonomy, BDHS 1999–00 to 2017–18.**

| Variable | | Unadjusted | | Adjusted[1] | |
|---|---|---|---|---|---|
| | | POR (95% CI) | p-value | POR (95% CI) | p-value |
| **Maternal age (Ref: 15–19 years)** | 20–29 | 1.6*** (1.4,1.7) | <0.001 | 1.3*** (1.2,1.4) | <0.001 |
| | 30–49 | 2.3*** (2.0,2.5) | <0.001 | 2.1*** (1.8,2.4) | <0.001 |
| **Number of children (Ref: 1)** | 2–3 | 1.6*** (1.5,1.7) | <0.001 | 1.4*** (1.3,1.6) | <0.001 |
| | 4 or More | 1.3*** (1.2,1.5) | <0.001 | 1.2** (1.1,1.4) | 0.004 |
| **Maternal education (Ref: No formal education)** | Primary | 1.2*** (1.1,1.3) | <0.001 | 1.2*** (1.1,1.4) | <0.001 |
| | Secondary | 1.3*** (1.2,1.4) | <0.001 | 1.4*** (1.2,1.6) | <0.001 |
| | College/above | 2.2*** (1.9,2.5) | <0.001 | 1.9*** (1.6,2.2) | <0.001 |
| **Paternal education (Ref: No formal education)** | | | | | |
| | Secondary | 1.2** (1.1,1.3) | 0.002 | 1.0 (0.9,1.1) | 0.62 |
| | College/above | 1.7*** (1.5,1.9) | <0.001 | 1.1 (0.9,1.3) | 0.28 |
| **Current work (Ref: No)** | Yes | 1.6*** (1.5,1.8) | <0.001 | 1.4*** (1.2,1.5) | <0.001 |
| **Religion (Ref: Muslim)** | Other | 1.0 (0.9,1.1) | 0.85 | | |
| **Mass media exposure (Ref: Not exposed)** | Exposed | 1.2*** (1.2,1.3) | < .001 | 1.1** (1.0,1.2) | 0.004 |
| **Wealth quintile (Ref: Poorest)** | Poorer | 1.0 (0.9,1.1) | 0.81 | 1.0 (0.9,1.1) | 0.44 |
| | Middle | 1.0 (0.9,1.2) | 0.49 | 0.9 (0.8,1.1) | 0.36 |
| | Richer | 1.2* (1.0,1.3) | 0.013 | 1.0 (0.8,1.1) | 0.48 |
| | Richest | 1.4*** (1.2,1.6) | < .001 | 1.0 (0.8,1.1) | 0.57 |
| **Place of residence (Ref: Urban)** | Rural | 0.7*** (0.6,0.8) | < .001 | 0.8*** (0.7,0.8) | <0.001 |
| **Division of residence (Ref: Dhaka)** | Chittagong | 0.9 (0.8,1.1) | 0.31 | 0.9 (0.8,1.0) | 0.18 |
| | Rajshahi | 1.0 (0.9,1.2) | 0.88 | 1.1 (0.9,1.3) | 0.24 |
| | Khulna | 0.9 (0.8,1.1) | 0.39 | 1.0 (0.8,1.1) | 0.6 |
| | Barisal | 0.8* (0.7,1.0) | 0.032 | 0.8* (0.7,1.0) | 0.028 |
| | Sylhet | 0.6*** (0.5,0.7) | <0.001 | 0.7*** (0.6,0.7) | <0.001 |
| | Rangpur | 1.2* (1.0,1.5) | 0.029 | 1.2 (1.0,1.5) | 0.063 |
| | Mymensingh | 2.0*** (1.4,2.7) | <0.001 | 1.2 (0.8,1.6) | 0.40 |
| **Survey year (Ref: 1999–00)** | 2004 | 0.8** (0.7,0.9) | 0.005 | 0.8*** (0.7,0.9) | <0.001 |
| | 2007 | 1.4*** (1.2,1.6) | <0.001 | 1.3*** (1.1,1.5) | <0.001 |
| | 2011 | 1.0 (0.9,1.1) | 0.76 | 0.9 (0.8,1.0) | 0.079 |
| | 2014 | 1.1 (0.9,1.2) | 0.43 | 0.9 (0.8,1.0) | 0.12 |
| | 2017–18 | 2.0*** (1.8,2.3) | <0.001 | 1.7*** (1.4,1.9) | <0.001 |

1. Adjusted for all variables in the column

Abbreviations: BDHS: Bangladesh Demographic & Health Survey; CI: Confidence interval; POR: Prevalence odds ratio

*: p<0.05

**: p<0.01

***: p<0.001

to socioeconomic/sociocultural differences between the districts of each division [7]. Future studies should explore the divisional differences and may include districts in the analysis.

Contrary to our hypothesis, paternal education and household wealth were not associated with maternal autonomy. Household wealth was computed by principal component analysis of basic household construction materials and household belongings and did not necessarily indicate maternal earning power or income, which may explain its lack of association [7]. We observed the positive association of current work with maternal autonomy; women in richer families may not need to work to earn money, that can negatively impact maternal autonomy [32]. Earlier studies showed similar results regarding the association with women's education

while lack of association with husbands' education or household wealth [11, 12, 33, 34]. On the other hand, the positive association of maternal autonomy with education and employment indicates the importance of 'individual' socioeconomic variables and of creating more educational and earning opportunities for mothers [12, 18].

There may not be direct measures or indicators for maternal autonomy [4], however, we used a simple definition with three variables present in every BDHS [7, 14–17, 35]. These variables provided insights about maternal autonomy in Bangladesh [7, 14–17, 35], and helped us to identify the mothers (e.g., low education or no work) who may have lower autonomy.

The positive association of maternal education or employment with their autonomy indicates that supporting community-based programs to enhance education levels or employment could increase maternal autonomy. This potential outcome may not be limited to individual empowerment and have broader implications for overall health, social dynamics, and economic development [10–13]. This could also reduce other disparities in healthcare utilization and outcomes. The absence of studies on trends/factors affecting maternal autonomy also signifies the importance of future quantitative (e.g., to understand the impact of other socioeconomic or sociocultural factors) and qualitative research (e.g., to understand the perspectives of husbands, in-laws, and other family members on the role of autonomy).

This study has several notable strengths. First, this was a large, nationally representative study covering rural and urban regions of all administrative divisions in Bangladesh. Covering all areas made these findings generalizable to the country. BDHS used a standardized and validated questionnaire that increased the authenticity of our findings. The response rate of the survey was high. This analysis was also conducted after accounting for all available confounders.

The limitations of the study also warrant discussion. First, we investigated the associations in multiple individual cross-sectional surveys, meaning that BDHS collected information on all variables simultaneously during each study wave. Therefore, any observed association between exposures and outcomes may not be causal. The data were collected based on self-reports, which may vary from actual decision-making, and there may be some overreporting to please the interviewers (i.e., social desirability bias) [6]. The data included up to three years after the birth; autonomy may differ from the period immediately after birth.

## Conclusions

Although maternal autonomy's prevalence has increased substantially during the past two decades in Bangladesh, a significant proportion of mothers still do not have any autonomy. In a patriarchal society like Bangladesh, where husbands are the primary decision-makers of families, it may require some time to improve the autonomy of mothers and change sociocultural norms, including attitudes towards maternal autonomy; however, our data suggest that it is possible to improve. Considering maternal autonomy's positive association with education and work, creating more educational or earning opportunities can increase maternal autonomy. More qualitative and quantitative studies are also required to understand how to improve it.

## Supporting information

**S1 Fig. Steps of data collection of Bangladesh Demographic and Health Survey 1999–2000, 2004, 2007, 2011, 2014, & 2017–18.**
(DOCX)

**S1 Table. Study variables.**
(DOCX)

**S2 Table. Prevalence (95% confidence interval) and trends of high maternal autonomy by sociodemographic characteristics.**
(DOCX)

**S3 Table. Comparison of study sample by presence of any maternal autonomy, BDHS 1999–00 to 2017–18, % (n).**
(DOCX)

**S4 Table. Comparison of study sample by level of autonomy, BDHS 1999–00 to 2017–18, % (n).**
(DOCX)

**S5 Table. Factors associated with maternal autonomy, BDHS 1999–00 to 2017–18.**
(DOCX)

## Acknowledgments

This manuscript was a part of the doctoral dissertation project of the primary author (i.e., Gulam Kibria). The primary author thanks his mentors, teachers, friends, family members, and other well-wishers for their support; he also thanks ICF International for the approval to use the data as well as the survey participants for their time. While no funding was received for this study, the author acknowledges the funding from the University of Maryland, Baltimore and Johns Hopkins University for his doctoral education

## Author Contributions

**Conceptualization:** Gulam Muhammed Al Kibria, Jennifer Albrecht, Jon Mark Hirshon.

**Formal analysis:** Gulam Muhammed Al Kibria.

**Investigation:** Gulam Muhammed Al Kibria, Jennifer Albrecht, Kristen A. Stafford, Laundette Jones, Roumen Vesselinov, Jon Mark Hirshon.

**Methodology:** Gulam Muhammed Al Kibria, Jennifer Albrecht, Wendy Lane, Kristen A. Stafford, Laundette Jones, Roumen Vesselinov, Jon Mark Hirshon.

**Project administration:** Jon Mark Hirshon.

**Supervision:** Jennifer Albrecht, Wendy Lane, Kristen A. Stafford, Laundette Jones, Roumen Vesselinov, Jon Mark Hirshon.

**Writing – original draft:** Gulam Muhammed Al Kibria.

**Writing – review & editing:** Jennifer Albrecht, Wendy Lane, Kristen A. Stafford, Laundette Jones, Roumen Vesselinov, Jon Mark Hirshon.

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
