## [Decision Letter · Decision Letter 0]

8 Nov 2023

PGPH-D-23-01928

Prevalence, Trends, and Factors Associated with Maternal Autonomy Regarding Healthcare, Finances, and Mobility in Bangladesh: Analysis of Demographic and Health Surveys 1999-2018

Dear Dr. Al Kibria,

Thank you for submitting your manuscript to PLOS Global Public Health. After careful consideration, we feel that it has merit but does not fully meet PLOS Global Public Health’s publication criteria as it currently stands. Therefore, we invite you to submit a revised version of the manuscript that addresses the points raised during the review process.

We look forward to receiving your revised manuscript.

Kind regards,

Zahra Zeinali, MD MPH DrGH (c)

Academic Editor

Journal Requirements:

Additional Editor Comments (if provided):

Reviewers' comments:

Reviewer's Responses to Questions

**Comments to the Author**

1. Does this manuscript meet PLOS Global Public Health’s publication criteria? Is the manuscript technically sound, and do the data support the conclusions? The manuscript must describe methodologically and ethically rigorous research with conclusions that are appropriately drawn based on the data presented.

Reviewer #1: Partly

Reviewer #2: Yes

2. Has the statistical analysis been performed appropriately and rigorously?

Reviewer #1: Yes

Reviewer #2: Yes

3. Have the authors made all data underlying the findings in their manuscript fully available (please refer to the Data Availability Statement at the start of the manuscript PDF file)?

Reviewer #1: Yes

Reviewer #2: Yes

4. Is the manuscript presented in an intelligible fashion and written in standard English?

Reviewer #1: Yes

Reviewer #2: Yes

5. Review Comments to the Author

Reviewer #1: Aside from various edits related to further explanation of points raised by the authors, the need for more narrative explanation of results and background research, as well as additional references to the specific context [in the conclusion], the draft requires slight adjustment to the conclusions presented. Following on from the stated hypothesis, ("the prevalence of maternal autonomy has increased and that women with higher socioeconomic status (e.g., education or wealth) have higher autonomy than their counterparts") the data displayed in Table 3, particularly relating to "wealth quintile" demonstrate that none of the adjusted figures are significantly associated with maternal autonomy. However, in your conclusion section (page 12), you state that your findings confirm your hypothesis. While I can take this to understand that the first part of the hypothesis related to a prevalence increase has been supported, the second part of the hypothesis has clearly been demonstrated to have mixed results. It is critical that this is adequately addressed.

Additionally, the inclusion of the data on "lone maternal autonomy" is not thoroughly addressed anywhere other than the brief section in the results. Please consider adding more context to these findings as they represent an important facet of your argument.

Reviewer #2: This manuscript reports on the trends in prevalence and factors associated with maternal autonomy using a common definition from DHS in Bangladesh. This manuscript is well-written and provides insight into how maternal autonomy might be defined; these findings could be of great interest to health economists and gender/health experts. A few comments to consider:

Abstract

It would be helpful to add some numbers to show the increase in maternal autonomy, rather than just " While the level of maternal autonomy has increased over the past two decades,..."

Introduction

Please add the number of surveys providing data (6?) when describing the overall study objective.

Methods

It would be helpful to state why only included women who had given birth within three years preceding the survey.

Other variables (page 5); please provide rationale for the other variables that were selected.

It would be useful to reference "freedom of mobility" when discussing the third variable described.

Results

What is the relationship between division of residence and place of residence? It seems that Dhaka and Chittagong would be the "urban" sample and the other districts would be "rural".

Discussion

I would love to see a larger discussion of regional differences in results and rationale for why that might be happening. Perhaps another area of future exploration would be to look at this issue within districts that were not included in DHS?

I question the idea that public campaigns will somehow be effective in addressing this long-standing issue of women's empowerment within a patriarchal social structure. The authors may consider removing the line from the discussion and/or justify why it might be a tenable way to address the issue at hand.

"More public campaigns should be carried out to increase awareness about maternal autonomy, education, and healthcare utilization/outcomes in Bangladesh".

Conclusion

The authors state that " it may require a long time to improve the autonomy of mothers and change sociocultural norms, ...". Given that they are reporting on an approximate 20 year period, I wonder if they might want to reframe this point in the positive. Twenty years for significant changes in women's empowerment seems like good progress to me.

6. PLOS authors have the option to publish the peer review history of their article (what does this mean?). If published, this will include your full peer review and any attached files.

**Do you want your identity to be public for this peer review?** For information about this choice, including consent withdrawal, please see our Privacy Policy.

Reviewer #1: No

Reviewer #2: No

---

## [Decision Letter · Decision Letter 1]

11 Dec 2023

PGPH-D-23-01928R1

Prevalence, Trends, and Factors Associated with Maternal Autonomy Regarding Healthcare, Finances, and Mobility in Bangladesh: Analysis of Demographic and Health Surveys 1999-2018

Dear Dr. Al Kibria,

Thank you for submitting your manuscript to PLOS Global Public Health. After careful consideration, we feel that it has merit but does not fully meet PLOS Global Public Health’s publication criteria as it currently stands. Therefore, we invite you to submit a revised version of the manuscript that addresses the points raised during the review process.

We look forward to receiving your revised manuscript.

Kind regards,

Zahra Zeinali, MD MPH DrGH (c)

Academic Editor

Journal Requirements:

2. Please provide separate figure files in .tif or .eps format only and remove any figures embedded in your manuscript file. Please also ensure all files are under our size limit of 10MB.

Additional Editor Comments (if provided):

Reviewers' comments:

Reviewer's Responses to Questions

**Comments to the Author**

1. If the authors have adequately addressed your comments raised in a previous round of review and you feel that this manuscript is now acceptable for publication, you may indicate that here to bypass the “Comments to the Author” section, enter your conflict of interest statement in the “Confidential to Editor” section, and submit your "Accept" recommendation.

Reviewer #1: All comments have been addressed

Reviewer #2: (No Response)

2. Does this manuscript meet PLOS Global Public Health’s publication criteria? Is the manuscript technically sound, and do the data support the conclusions? The manuscript must describe methodologically and ethically rigorous research with conclusions that are appropriately drawn based on the data presented.

Reviewer #1: Yes

Reviewer #2: (No Response)

3. Has the statistical analysis been performed appropriately and rigorously?

Reviewer #1: Yes

Reviewer #2: (No Response)

4. Have the authors made all data underlying the findings in their manuscript fully available (please refer to the Data Availability Statement at the start of the manuscript PDF file)?

Reviewer #1: Yes

Reviewer #2: (No Response)

5. Is the manuscript presented in an intelligible fashion and written in standard English?

Reviewer #1: Yes

Reviewer #2: (No Response)

6. Review Comments to the Author

Reviewer #1: Please see the attached document

Reviewer #2: (No Response)

7. PLOS authors have the option to publish the peer review history of their article (what does this mean?). If published, this will include your full peer review and any attached files.

**Do you want your identity to be public for this peer review?** For information about this choice, including consent withdrawal, please see our Privacy Policy.

Reviewer #1: No

Reviewer #2: No

---

## [Editor Report · Decision Letter 2]

27 Dec 2023

Prevalence, Trends, and Factors Associated with Maternal Autonomy Regarding Healthcare, Finances, and Mobility in Bangladesh: Analysis of Demographic and Health Surveys 1999-2018

PGPH-D-23-01928R2

Dear Dr. Al Kibria,

We are pleased to inform you that your manuscript 'Prevalence, Trends, and Factors Associated with Maternal Autonomy Regarding Healthcare, Finances, and Mobility in Bangladesh: Analysis of Demographic and Health Surveys 1999-2018' has been provisionally accepted for publication in PLOS Global Public Health.

Best regards,

Zahra Zeinali, MD MPH DrGH (c)

Academic Editor
